# Clustering-friendly Representation Learning via Instance Discrimination and Feature Decorrelation

**Yaling Tao, Kentaro Takagi & Kouta Nakata**
Corporate R&D Center, Toshiba Corporation
1, Komukai Toshiba-cho, Saiwai-ku, Kawasaki, Kanagawa, Japan
`{yaling1.tao,kentaro1.takagi,kouta.nakata}@toshiba.co.jp`

## Abstract

Clustering is one of the most fundamental tasks in machine learning. Recently, deep clustering has become a major trend in clustering techniques. Representation learning often plays an important role in the effectiveness of deep clustering, and thus can be a principal cause of performance degradation. In this paper, we propose a clustering-friendly representation learning method using instance discrimination and feature decorrelation. Our deep-learning-based representation learning method is motivated by the properties of classical spectral clustering. Instance discrimination learns similarities among data and feature decorrelation removes redundant correlation among features. We utilize an instance discrimination method in which learning individual instance classes leads to learning similarity among instances. Through detailed experiments and examination, we show that the approach can be adapted to learning a latent space for clustering. We design novel softmax-formulated decorrelation constraints for learning. In evaluations of image clustering using CIFAR-10 and ImageNet-10, our method achieves accuracy of $81.5\%$ and $95.4\%$, respectively. We also show that the softmax-formulated constraints are compatible with various neural networks.

## 1 Introduction

Clustering is one of the most fundamental tasks in machine learning. Recently, deep clustering has become a major trend in clustering techniques. In a fundamental form, autoencoders are used for feature extraction, and classical clustering techniques such as $k$-means are serially applied to the features. Recent deep clustering techniques integrate learning processes of feature extraction and clustering, yielding high performance for large-scale datasets such as handwritten digits Hu et al. (2017); Shaham et al. (2018); Xie et al. (2016); Tao et al. (2018). However, those methods have fallen short when targets become more complex, as in the case of real-world photograph dataset CIFAR-10 Krizhevsky et al. (2009). Several works report powerful representation learning leads to improvement of clustering performance on complex datasets Chang et al. (2017); Wu et al. (2019). Learning representation is a key challenge to unsupervised clustering.

In order to learn representations for clustering, recent works utilize metric learning which automatically learns similarity functions from data Chang et al. (2017); Wu et al. (2019). They assign pseudo-labels or pseudo-graph to unlabeled data by similarity measures in latent space, and learn discriminative representations to cluster data. These works improve clustering performance on real world images such as CIFAR-10 and ImageNet-10, and indicate the impact of representation learning on clustering. Although features from learned similarity function and pseudo-labels work well for clustering, algorithms still seem to be heuristic; we design a novel algorithm which is based on knowledge from established clustering techniques. In this work, we exploit a core idea of spectral clustering which uses eigenvectors derived from similarities.

Spectral clustering has been theoretically and experimentally investigated, and known to outperform other traditional clustering methods Von Luxburg (2007). The algorithm involves similarity matrix construction, transformation from similarity matrix to Laplacian, and eigendecomposition. Based on

eigenvectors, data points are mapped into a lower dimensional representation which carries information of similarities and is preferable for clustering. We bring this idea of eigenvector representation into deep representation learning.

We design the representation learning with two aims: 1) learning similarities among instances; and 2) reducing correlations within features. The first corresponds to Laplacian, and the second corresponds to feature orthogonality constrains in the spectral clustering algorithm. Learning process integrating both is relevant to eigendecomposition of Laplacian matrix in the spectral clustering.

For the first aim, we adopt the instance discrimination method presented in Wu et al. (2018), where each unlabeled instance is treated as its own distinct class, and discriminative representations are learned to distinguish between individual instance classes. This numerous-class discriminative learning enables learning partial but important features, such as small foreground objects in natural images. Wu et al. (2018) showed that the representation features retain apparent similarity among images and improve the performance of image classification by the nearest neighbor method. We extend their work to the clustering tasks. We clarify their softmax formulation works like similarity matrix in spectral clustering under the condition that temperature parameter $\tau$, which was underexplored in Wu et al. (2018), is set to be a larger value .

For the second aim, we introduce constraints which have the effect of making latent features orthogonal. Orthogonality is often an essential idea in dimension reduction methods such as principal components analysis, and it is preferable for latent features to be independent to ensure that redundant information is reduced. Orthogonality is also essential to a connection between proposed method and spectral clustering, as stated in Section 3.4. In addition to a simple soft orthogonal constraint, we design a novel softmax-formulated decorrelation constraint. Our softmax constraint is "softer" than the soft orthogonal constraint for learning independent feature spaces, but realizes stable improvement of clustering performance.

Finally, we combine instance discrimination and feature decorrelation into learning representation to improve the performance of complex image clustering. For the CIFAR-10 and ImageNet-10 datasets, our method achieves accuracy of $81.5\%$ and $95.4\%$, respectively. Our PyTorch Paszke et al. (2019) implementation of IDFD is available at `https://github.com/TTN-YKK/Clustering_friendly_representation_learning`.

Our main contributions are as follows:

- We propose a clustering-friendly representation learning method combining instance discrimination and feature decorrelation based on spectral clustering properties.

- We adapt deep representation learning by instance discrimination to clustering and clarify the essential properties of the temperature parameter.

- We design a softmax-formulated orthogonal constraint for learning latent features and realize stable improvement of clustering performance.

- Our representation learning method achieves performance comparable to state-of-the-art levels for image clustering tasks with simple $k$-means.

## 2 RELATED WORK

Deep clustering methods offer state-of-the-art performance in various fields. Most early deep clustering methods, such as Vincent et al. (2010); Tian et al. (2014), are two-stage methods that apply clustering after learning low-dimensional representations of data in a nonlinear latent space. The autoencoder method proposed in Hinton & Salakhutdinov (2006) is one of the most effective methods for learning representations. Recent works have simultaneously performed representation learning and clustering Song et al. (2013); Xie et al. (2016); Yang et al. (2017); Guo et al. (2017); Tao et al. (2018). Several methods based on generative models have also been proposed Jiang et al. (2016); Dilokthanakul et al. (2016). These methods outperform conventional methods, and sometimes offer performance comparable to that of supervised learning for simple datasets. Deep-learning-based unsupervised image clustering is also being developed Chang et al. (2017); Wu et al. (2019); Ji et al. (2019); Gupta et al. (2020); Van Gansbeke et al. (2020).

Several approaches focus on learning discriminative representations via deep learning. Bojanowski & Joulin (2017) found a mapping between images on a uniformly discretized target space, and enforced their representations to resemble a distribution of pairwise relationships. Caron et al. (2018) applied pseudo-labels to output as supervision by $k$-means and then trained a deep neural network. Donahue et al. (2016) proposed bidirectional generative adversarial networks for learning generative models that map simple latent distributions to complex real distributions, in order for generators to capture semantic representations. Hjelm et al. (2018) proposed deep infomax to maximize mutual information between the input and output of an encoder. Wu et al. (2018) was motivated by observations in supervised learning that the probabilities of similar image classes become simultaneously high. They showed that discriminating individual instance classes leads to learning representations that retain similarities among data.

IIC Ji et al. (2019) and SCAN Van Gansbeke et al. (2020) are two recent works focusing on image clustering and obtained high performance. IIC Ji et al. (2019) directly learns semantic labels without learning representations based on mutual information between image pairs. SCAN Van Gansbeke et al. (2020) focuses on the clustering phase and largely improved performance based on a given pre-designed representation learning. By contrast, we focus on learning a clustering-friendly representation space where objects can be simply clustered.

Our method exploits the idea of spectral clustering Shi & Malik (2000); Meila & Shi (2001); Von Luxburg (2007); Ng et al. (2002). From one perspective, spectral clustering finds a low dimensional embedding of data in the eigenspace of the Laplacian matrix, which is derived from pairwise similarities between data. By using the embedded representations, we can proceed to cluster the data by the $k$-means algorithm in the low-dimensional space. Spectral clustering often outperforms earlier algorithms such as $k$-means once pair similarities are properly calculated. Shaham et al. (2018) incorporated the concept of spectral clustering into deep a neural network structure. Similarities were calculated by learning a Siamese net Shaham & Lederman (2018) where the input positive and negative pairs were constructed according to the Euclidean distance.

## 3 PROPOSED METHOD

Given an unlabeled dataset $X = \{x_i\}_{i=1}^n$ and a predefined number of clusters $k$, where $x_i$ denotes the $i$th sample, we perform the clustering task in two phases, namely, representation learning and clustering. This work focuses on the first phase, which aims to learn an embedding function $\boldsymbol{v} = f_\theta(x)$ mapping data $\boldsymbol{x}$ to representation $\boldsymbol{v}$ so that $\boldsymbol{v}$ is preferable for clustering. $f_\theta$ is modeled as a deep neural network with parameter $\boldsymbol{\theta}$. We use $V = \{v_i\}_{i=1}^n$ to denote the whole representation set.

### 3.1 INSTANCE DISCRIMINATION

We apply the instance discrimination method proposed by Wu et al. (2018) to learn clustering-friendly representations that capture similarity between instances. The objective function is formulated based on the softmax criterion. Each instance is assumed to represent a distinct class. For given data $x_1, \ldots, x_n$, the corresponding representations are $\boldsymbol{v_1}, \ldots, \boldsymbol{v_n}$, and data $x_i$ is classified into the $i$th class. Accordingly, the weight vector for the $i$th class can be approximated by a vector $\boldsymbol{v_i}$. The probability of representation $\boldsymbol{v}$ being assigned into the $i$th class is

$$P(i|\boldsymbol{v}) = \frac{\exp(\boldsymbol{v_i}^T \boldsymbol{v}/\tau)}{\sum_{j=1}^n \exp(\boldsymbol{v_j}^T \boldsymbol{v}/\tau)}, \tag{1}$$

where $\boldsymbol{v_j}^T \boldsymbol{v}$ measures how well $\boldsymbol{v}$ matches the $j$th class, $\tau$ is a temperature parameter that controls the concentration of the distribution Hinton et al. (2015), and $\boldsymbol{v}$ is normalized to $||\boldsymbol{v}|| = 1$.

The objective maximizes the joint probability $\prod_{i=1}^n P_\theta(i|f_\theta(x_i))$ as

$$L_I = -\sum_{i=1}^n \log P(i|f_\theta(x_i)) = -\sum_i^n \log\left(\frac{\exp(\boldsymbol{v_i}^T \boldsymbol{v_i}/\tau)}{\sum_{j=1}^n \exp(\boldsymbol{v_j}^T \boldsymbol{v_i}/\tau)}\right). \tag{2}$$

Wu et al. (2018) shows that features obtained by minimizing the objective retain similarity between image instances and improve the performance of nearest neighbor classification. For clustering, we note that the parameter $\tau$, which is underexplored in Wu et al. (2018), has a large impact on clustering performance. The effect of $\tau$ is discussed later and experimental results are shown in 4.2.1.

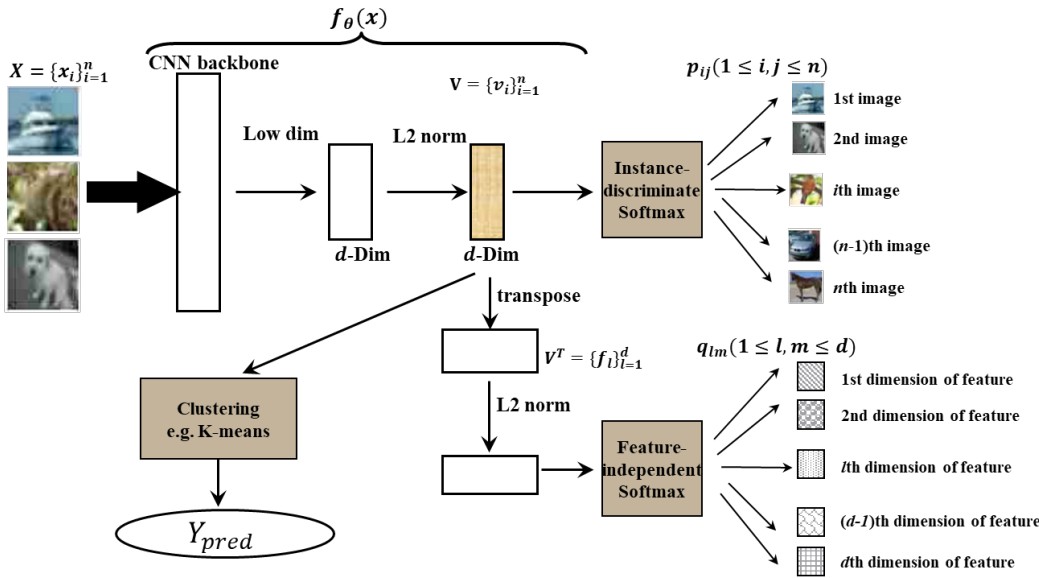

Figure 1: Pipeline of our method.

## 3.2 FEATURE DECORRELATION

We define a set of latent feature vectors $\boldsymbol{f}$ and use $f_l$ to denote the $l$th feature vector. Transposition of latent vectors $V$ coincides with $\{f_l\}_{l=1}^d$, where $d$ is the dimensionality of representations.

The simple constraint for orthogonal features is,

$$L_{FO} = ||VV^T - I||^2 = \sum_{l=1}^d \left( (\boldsymbol{f}_l^T \boldsymbol{f}_l - 1)^2 + \sum_{j=1, j \neq l}^n (\boldsymbol{f}_j^T \boldsymbol{f}_l)^2 \right). \tag{3}$$

Our novel constraint is based on a softmax formulation of

$$Q(l|\boldsymbol{f}) = \frac{\exp(\boldsymbol{f}_l^T \boldsymbol{f}/\tau_2)}{\sum_{m=1}^d \exp(\boldsymbol{f}_m^T \boldsymbol{f}/\tau_2)}, \tag{4}$$

$Q(l|\boldsymbol{f})$ is analogous to $P(i|\boldsymbol{v})$. $Q(l|\boldsymbol{f})$ measures how correlated a feature vector is to itself and how dissimilar it is to others. $\tau_2$ is the temperature parameter. We formulate the feature decorrelation constraint as

$$L_F = -\sum_{l=1}^d \log Q(l|\boldsymbol{f}) = \sum_{l=1}^d \left( -\boldsymbol{f}_l^T \boldsymbol{f}_l/\tau_2 + \log \sum_j^d \exp(\boldsymbol{f}_j^T \boldsymbol{f}_l/\tau_2) \right). \tag{5}$$

Both constrains in Eq. (3) and Eq. (5) aim to construct independent features. Conventionally, it is preferable for features to be independent to ensure that redundant information is reduced, and orthogonality is a common technique. Compare Eq. (3) and Eq. (5), we can see that minimizing $L_F$ and $L_{FO}$ can result in a similar effect, $f_l^T f_l \to 1$ and $f_j^T f_l \to -1$ or $0(l \neq j)$, and both try to decorrelate latent features.

Our softmax constraint in Eq. (5) shows practical advantages in flexibility and stability. Eq. (3) is called a soft orthogonal constraint, but is still strict enough to force the features to be orthogonal. If $d$ is larger than underlying structures that are hidden and unknown, all features are forcibly orthogonalized and the resultant features may not be appropriate. Softmax formulation allows off-diagonal elements to be non-zero and alleviates the problem of strict orthogonality.

Partial derivatives of $L_F$ and $L_{FO}$ with respect to $z_{jl} = f_j^T f_l$ are calculated as $\frac{\partial L_F}{\partial z_{jl}} = -\frac{1}{\tau_2} \delta_{jl} + \frac{1}{\tau_2} \frac{\exp(z_{jl}/\tau_2)}{\sum_j^d \exp(z_{jl}/\tau_2)}$ and $\frac{\partial L_{FO}}{\partial z_{jl}} = -2\delta_{jl} + 2z_{jl}$, where $\delta_{jl}$ is an indicator function. Since the derivatives

nearly equal zero due to $z_{jl} = 1$ in the case of $j = l$, we focus on the case of $j \neq l$. When $j \neq l$, the ranges of partial derivatives are $0 \leq \frac{\partial L_F}{\partial z_{jl}} \leq \frac{1}{\tau_2}$ and $-2 \leq \frac{\partial L_{FO}}{\partial z_{jl}} \leq 2$. The monotonicity of $L_F$ can lead to more stable convergence. The advantages of $L_F$ are confirmed by experiments in section 4.

### 3.3 Objective Function and Learning Model

Combining instance discrimination and feature decorrelation learning, we formulate our objective function $L_{IDFD}$ as follows:

$$L_{IDFD} = L_I + \alpha L_F, \tag{6}$$

Where $\alpha$ is a weight that balances the contributions of two terms $L_I$ and $L_F$.

Figure 1 shows the learning process for the motif of image clustering. Input images $X$ are converted into feature representations $V$ in a lower $d$-dimensional latent space, via nonlinear mapping with deep neural networks such as ResNet He et al. (2016). The $d$-dimensional vectors are simultaneously learned through instance discrimination and feature decorrelation. A clustering method, such as classical $k$-means clustering, is then used on the learned representations to obtain the clustering results.

Optimization can be performed by mini-batch training. To compute the probability $P(i|\boldsymbol{v})$ in Eq. (1), $\{\boldsymbol{v}_j\}$ is needed for all images. Like Wu et al. (2018); Xiao et al. (2017), we maintain a feature memory bank for storing them. For $Q(l|\boldsymbol{f})$ in Eq. (4), all $\{\boldsymbol{f}_m\}$ of $d$ dimensions in the current mini-batch can be obtained, we simply calculate the $Q(l|\boldsymbol{f})$ within the mini-batches.

We combine $L_I$ and $L_{FO}$ to formulate an alternative loss $L_{IDFO}$ in E.q. (7),

$$L_{IDFO} = L_I + \alpha L_{FO}. \tag{7}$$

We refer to representation learning using $L_{IDFD}$, $L_{IDFO}$, and $L_I$ loss as instance discrimination and feature decorrelation (IDFD), instance discrimination and feature orthogonalization (IDFO), and instance discrimination (ID), respectively.

### 3.4 Connection with Spectral Clustering

We explain the connection between IDFD and spectral clustering. We consider a fully connected graph consisting of all representation points, and the similarity matrix $W$ and degree matrix $D$ can be written as $W_{ij} = \exp(v_i^T v_j / \tau)$ and $D_{ii} = \sum_m^n \exp(v_i^T v_m / \tau)$. The loss function of spectral clustering Shaham et al. (2018) can be reformulated as

$$L_{SP} = (Tr)(\boldsymbol{f}L\boldsymbol{f}) = \frac{1}{2}\sum_k \sum_{ij}^n w_{ij}(f_i^k - f_j^k)^2 = \frac{1}{2}\sum_k \sum_{ij}^n \exp\left(\frac{v_i^T v_j}{\tau}\right)||v_i - v_j||^2, \tag{8}$$

where $L$ is Laplacian matrix, $\boldsymbol{f}$ are feature vectors. Spectral clustering is performed by minimizing $L_{SP}$ subject to orthogonal condition of $\boldsymbol{f}$, and when $L_{SP}$ takes minimum value $\boldsymbol{f}$ become eigenvectors of Laplacian $L$. According to Section 3.2, minimizing $L_F$ can approximate the orthogonal condition. Under this condition, minimizing $L_I$ can approximate the minimizing $L_{SP}$, which is explained as follows.

According to Eq.(2), minimizing loss $L_I$ means maximizing $v_i^T v_i$ and minimizing $v_i^T v_j$. When $i = j$, we have $||v_i - v_j||^2 = 0$, $L_{SP}$ becomes zero. We need consider only the influence on $L_{SP}$ from minimizing $v_i^T v_j$. As $\boldsymbol{v}$ are normalized, $L_{SP}$ can be rewritten using cosine metric as

$$L_{SP} = \sum_{ij}^n \exp\left(\frac{\cos\theta}{\tau}\right)\sin^2\frac{\theta}{2}, \tag{9}$$

then $\frac{\partial L_{SP}}{\partial \theta}$ can be calculated as

$$\frac{\partial L_{SP}}{\partial \theta} = \frac{1}{\tau}\sin\theta(\tau - 1 + \cos\theta)\exp\left(\frac{\cos\theta}{\tau}\right). \tag{10}$$

According to Eq.(10), we get $\frac{\partial L_{SP}}{\partial \theta} \geq 0$ when $\tau \geq 2$. This means $L_{SP}$ monotonically decreases when we minimize $v_i^T v_j$. Therefore, the impact from minimizing $v_i^T v_j$ is good for minimizing $L_{SP}$. Even if $\tau$ is a little smaller than 2, because $\tau$ controls the scale of derivatives and the range of $\theta$ where the derivative is negative, large $\tau$ decreases the scale and narrows the range, resulting in a small influence on the total loss. From this viewpoint, the effectiveness of minimizing $L_I$ using large $\tau$ is approximately the same as that of $L_{SP}$. By adding feature decorrelation constraints, IDFD becomes analogous to spectral clustering.

## 4 EXPERIMENTS

We conducted experiments using five datasets: **CIFAR-10** Krizhevsky et al. (2009), **CIFAR-100** Krizhevsky et al. (2009), **STL-10** Coates et al. (2011), **ImageNet-10** Deng et al. (2009), and **ImageNet-Dog** Deng et al. (2009). We adopted ResNet18 He et al. (2016) as the neural network architecture in our main experiments. The same architecture is used for all datasets. Our experimental settings are in accordance with that of Wu et al. (2018). Data augmentation strategies often used on images are also adopted in experiments. Details about datasets and experimental setup are given in Appendix A.

For IDFD, the weight $\alpha$ is simply fixed at 1. Orthogonality constraint weights for IDFO were $\alpha = 10$ on CIFAR-10 and CIFAR-100, and $\alpha = 0.5$ on STL-10 and ImageNet subsets. The weight $\alpha$ was set according to the orders of magnitudes of losses. In the main experiments, we set temperature parameter $\tau = 1$ for IDFO and IDFD, and $\tau_2 = 2$ for IDFD. In order to fully investigate our work, we also constructed two versions of instance discrimination (ID) that uses only $L_I$ loss, ID(original) with small $\tau = 0.07$ and ID(tuned) with large $\tau = 1$.

We compared ID(tuned), IDFO, and IDFD with ID(original) and six other competitive methods, clustering with an autoencoder (AE) Hinton & Salakhutdinov (2006), deep embedded clustering (DEC) Xie et al. (2016), deep adaptive image clustering (DAC) Chang et al. (2017), deep comprehensive correlation mining (DCCM) Wu et al. (2019), invariant information clustering (IIC) Ji et al. (2019), and semantic clustering by adopting nearest neighbors (SCAN) Van Gansbeke et al. (2020) .We use three metrics to measure clustering performance: standard clustering accuracy (ACC), normalized mutual information (NMI), and adjusted rand index (ARI). These metrics give values in $[0, 1]$, with higher scores indicating more accurate clustering assignments.

### 4.1 MAIN RESULTS

Table 1 lists the best performances for each method. The results for the four methods AE, DEC, DAC, and DCCM are cited from Wu et al. (2019), and results for two methods IIC and SCAN are cited from Van Gansbeke et al. (2020). Comparing these results, we conclude that ID(tuned), IDFO, and IDFD, clearly outperform these methods excluding SCAN for all datasets, according to the metrics ACC, NMI, and ARI. For dataset CIFAR-10, ID(tuned), IDFO, and IDFD yielded ACC values of $77.6\%$, $82.8\%$, and $81.5\%$, respectively. For dataset ImageNet-10, ID(tuned), IDFO, and IDFD achieved ACC values of $93.7\%$, $94.2\%$, and $95.4\%$. The high performance is comparable with that of supervised and semi-supervised methods. Gaps between the results of ID(tuned) and those of IDFO and IDFD reflect the effect of the feature constraint term. The performance is improved for all datasets by introducing feature orthogonalization and decorrelation. Impressively, ID(tuned) significantly outperformed ID(original) on all datasets, showing strong impact of temperature parameter. This will be discussed separately in section 4.2.1.

In addition, we note that IDFD differs from SCAN in that IDFD focuses on the representation leaning while SCAN focuses on clustering by given a representation learning. Both SCAN and IDFD demonstrate significant improvement on performance compared with other methods. Results of IDFD and SCAN showed effectiveness of efforts on both representation learning and clustering phases of deep clustering.

We also examine the learning stability of ID(tuned), IDFO, and IDFD. Figure 2 illustrates the accuracy on CIFAR-10 running each of ID(tuned), IDFO, and IDFD. We can see that both IDFO and IDFD obtained higher peak ACC values than ID(tuned). In particular, IDFD yielded higher performance than ID over the entire learning process. IDFO performed better than the other two methods and obtained the highest ACC value in earlier epochs. However, the ACC widely fluctuated

Table 1: Clustering results (%) of various methods on five datasets.

| Dataset | CIFAR-10 | | | CIFAR-100 | | | STL-10 | | | ImageNet-10 | | | ImageNet-Dog | | |
|---|---|---|---|---|---|---|---|---|---|---|---|---|---|---|---|
| Metric | ACC | NMI | ARI | ACC | NMI | ARI | ACC | NMI | ARI | ACC | NMI | ARI | ACC | NMI | ARI |
| AE | 31.4 | 23.9 | 16.9 | 16.5 | 10.0 | 4.8 | 30.3 | 25.0 | 16.1 | 31.7 | 21.0 | 15.2 | 18.5 | 10.4 | 7.3 |
| DEC | 30.1 | 25.7 | 16.1 | 18.5 | 13.6 | 5.0 | 35.9 | 27.6 | 18.6 | 38.1 | 28.2 | 20.3 | 19.5 | 12.2 | 7.9 |
| DAC | 52.2 | 39.6 | 30.6 | 23.8 | 18.5 | 8.8 | 47.0 | 36.6 | 25.7 | 52.7 | 39.4 | 30.2 | 27.5 | 21.9 | 11.1 |
| DCCM | 62.3 | 49.6 | 40.8 | 32.7 | 28.5 | 17.3 | 48.2 | 37.6 | 26.2 | 71.0 | 60.8 | 55.5 | 38.3 | 32.1 | 18.2 |
| ID(original) | 44.0 | 30.9 | 22.1 | 26.7 | 22.1 | 10.8 | 51.4 | 36.2 | 28.5 | 63.2 | 47.8 | 42.0 | 36.5 | 24.8 | 17.2 |
| IIC | 61.7 | 51.1 | 41.1 | 25.7 | 22.5 | 11.7 | 59.6 | 49.6 | 39.7 | - | - | - | - | - | - |
| SCAN | 88.3 | 79.7 | 77.2 | 50.7 | 48.6 | 33.3 | 80.9 | 69.8 | 64.6 | - | - | - | - | - | - |
| ID(tuned) | 77.6 | 68.2 | 61.6 | 40.9 | 39.2 | 24.3 | 72.6 | 64.0 | 52.6 | 93.7 | 86.7 | 86.5 | 47.6 | 47.0 | 33.5 |
| IDFO | 82.8 | 71.4 | 67.9 | 42.5 | 43.2 | 24.4 | 75.6 | 63.6 | 56.9 | 94.2 | 87.1 | 87.6 | 61.2 | 57.9 | 41.4 |
| IDFD | 81.5 | 71.1 | 66.3 | 42.5 | 42.6 | 26.4 | 75.6 | 64.3 | 57.5 | 95.4 | 89.8 | 90.1 | 59.1 | 54.6 | 41.3 |

over the learning process and dropped in later epochs. As analyzed in 3.2, our proposed IDFD makes performance higher than ID and more stable than IDFO.

## 4.2 DISCUSSION

### 4.2.1 ANALYSIS ON TEMPERATURE PARAMETER

Gaps between results of ID(original) and ID(tuned) in Table 1 show strong impact of temperature parameter. We theoretically and intuitively analyze the essential change caused by the temperature parameter in this subsection.

First, we consider why instance-level discrimination works and under what conditions. Difference in the performance of ID(original) and ID(tuned) suggests optimal distribution in latent space changes with the magnitude of $\tau$. According to empirical investigation and theoretical analysis, we find that a large $\tau$ in $L_I$ encourages data points to follow a compact distribution when minimizing the loss, while a small $\tau$ drives them to follow a uniform distribution. This means minimizing $L_I$ with a large $\tau$ can reach a good clustering-friendly solution. This property was explained by demonstrating examples and calculation, details are given in Appendix B.

In the definition of $P(i|\boldsymbol{v})$ in Eq. (1), when $\tau$ is small, we compute softmax on larger logits, resulting in higher prediction, and obtain a more confident model. From this viewpoint, we can leverage a small $\tau$ to decrease class entanglement if we can learn an accurate class-weight vector. In the general classification problem, since the weight of each class can be learned according to the real labels, it is preferable for models to be more confident. Most works therefore recommend setting a small value, such as $\tau = 0.07$ Wu et al. (2018). In clustering, however, instance-level discrimination is used to learn similarity among samples, with only one sample in each class. Because the model is highly confident, each sample tends to be completely independent from each other. Similarity among samples is seemingly encouraged to approach close to zero, even for samples from the same class. This clearly deviates from the original intent of adopting instance-level discrimination to learn sample entanglements under the condition that each sample can be discriminative. A larger $\tau$ than that used for classification is thus needed.

More experiments over different temperature settings on ID and IDFD were conducted on CIFAR-10. Figure 3 shows the accuracy of ID for $\tau = \{0.07, 0.2, 0.5, 0.8, 1, 2, 5, 10\}$. We calculated the mean and standard deviation of ACC values over the last 500 epochs for each experiment. From the results, we can see that ID can suffer significant performance degradation when $\tau$ is too small or too large. This agrees with our analysis above. We also investigate the impact of $\tau_2$ by fixing $\tau = 1$. Figure 4 shows the accuracy of the IDFD for $\tau_2 = \{0.1, 0.5, 1, 2, 3, 4, 5, 10\}$. Experimental results show that IDFD is relatively robust to the parameter $\tau_2$ and enables stable representation learning.

### 4.2.2 REPRESENTATION DISTRIBUTION AND FEATURE BEHAVIOR

Figure 5 visualizes the results of representations learned in four experiments: (a) ID(original), (b) ID(tuned), (c) IDFO with $\tau = 1$ and $\alpha = 10$, and (d) IDFD with $\tau = 1$, $\tau_2 = 2$, and $\alpha = 1$ on CIFAR-10. 128-dimension representations were embedded into two dimensions by t-SNE (t-distributed stochastic neighbor embedding) Maaten & Hinton (2008). Colors indicate ground truth classes. The distributions for the ID(original) and ID(tuned) again show the significant difference between

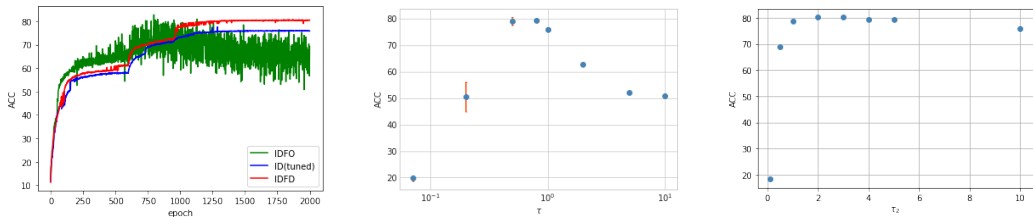

Figure 2: ACC values over learning process.

Figure 3: Accuracy of ID for various $\tau$ settings.

Figure 4: Accuracy of IDFD for various $\tau_2$ settings.

them. Data distribution when $\tau = 1$ is apparently more clustering-friendly than when $\tau = 0.07$. Furthermore, compared with ID(tuned), IDFO and IDFD can separate samples from different classes with certain margins. IDFO tended to construct a patch-like distribution within one class. In contrast, IDFD maintained a tighter connection among samples of the same class and more distinct borders between different classes.

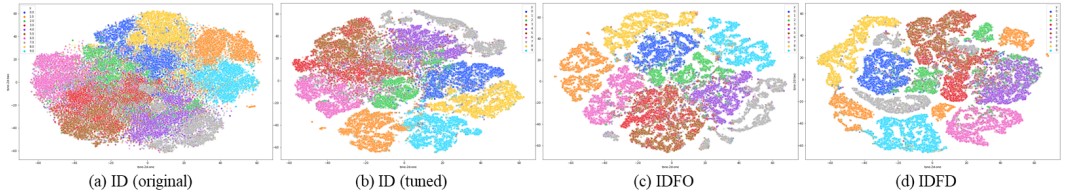

(a) ID (original)        (b) ID (tuned)        (c) IDFO        (d) IDFD

Figure 5: Distribution of feature representations on CIFAR-10.

Figure 6 shows distribution of feature representations on ImageNet-10 learned by IDFD. We can see that representations of ImageNet-10 are clustering-friendly and even better than that of CIFAR-10. This is consistent with the results in Table 1 evaluated by metrics ACC, NMI, and ARI. In addition to that, we also plot sample images corresponding to points lying near the border between clusters. We can see that these samples are certainly similar in appearance.

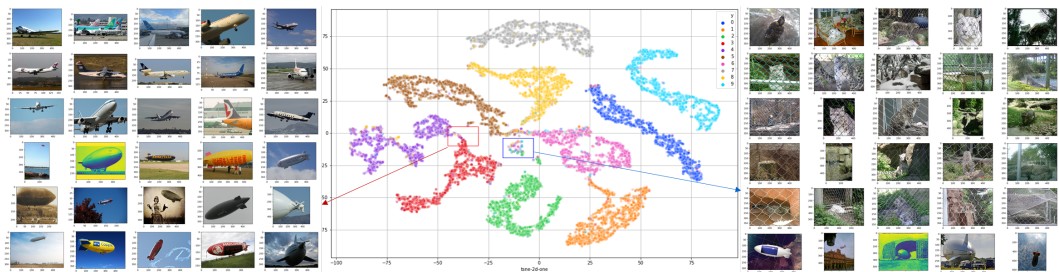

Figure 6: Distribution of feature representations on ImageNet-10 learned by IDFD and samples corresponding to points in some areas.

We investigate the effects of orthogonal and decorrelation constraints $L_{FO}$ and $L_F$. Figure 7 illustrates the feature correlations of ID(tuned), IDFO, and IDFD on dataset CIFAR-10. We see that IDFO clearly decorrelates features and IDFD retains a moderate level of feature correlation between ID and IDFD. Taken together with Figure 2, these results suggest that the softmax formulation of IDFD alleviates the problem of strict orthogonality and enables stable representation learning.

### 4.2.3 INVESTIGATION FOR PRACTICAL USE

We investigate the dependencies of our method on networks through experiments on other networks: ConvNet Wu et al. (2019), VGG16 Simonyan & Zisserman (2014), and ResNet34 He et al. (2016). Performance was evaluated using the CIFAR-10 dataset. Results listed in Table 2 show that IDFD

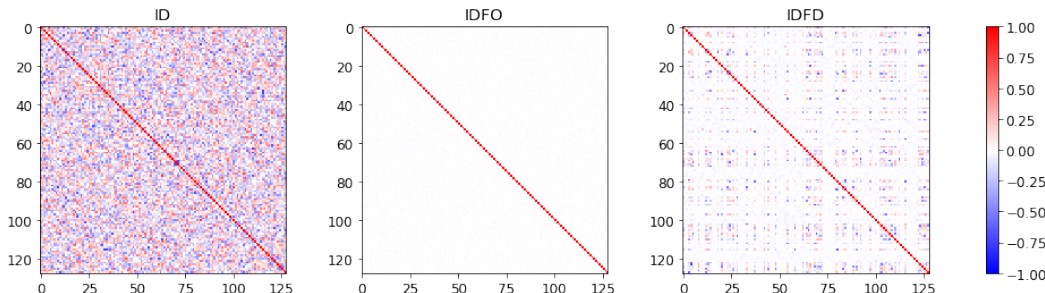

Figure 7: Feature correlation matrix on CIFAR-10 with ResNet18

can work on various networks. IDFD outperforms ID(tuned), and FD term shows more obvious effect on these networks. We also confirm the effect of cooperation between $L_I$ and $L_F$ from the viewpoint of spectral clustering, combinations of AE and $L_F$ were evaluated in terms of clustering performance. We found that AE cannot benefit from $L_F$ as $L_I$ did. This result verified that $L_F$ has a deep relation with $L_I$, and IDFD is not a simple combination. We also investigate the importance of data augmentation in performance through experiments. Due to the page limit, our extended experiments are given in Appendix C.

Table 2: Clustering results (%) on various network architectures.

| Network | ConvNet | | | VGG16 | | | ResNet18 | | | ResNet34 | | |
|---------|------|------|------|------|------|------|------|------|------|------|------|------|
| Metric | ACC | NMI | ARI | ACC | NMI | ARI | ACC | NMI | ARI | ACC | NMI | ARI |
| ID(tuned) | 26.8 | 15.0 | 8.9 | 39.3 | 31.6 | 20.9 | 77.6 | 68.2 | 61.6 | 80.2 | 71.1 | 64.6 |
| IDFD | 42.0 | 32.7 | 23.2 | 56.8 | 46.7 | 36.5 | 81.5 | 71.1 | 66.3 | 82.7 | 73.4 | 68.4 |

## 5 CONCLUSION

We present a clustering-friendly representation learning method combining instance discrimination and feature decorrelation based on spectral clustering properties. Instance discrimination learns similarities among data and feature decorrelation removes redundant correlation among features. We analyzed why instance discrimination works for clustering and clarified the conditions. We designed a softmax-formulated feature decorrelation constraint for learning the latent space to realize stable improvement of clustering performance. We also explained the connection between our method and spectral clustering. The proposed representation learning method achieves accuracies comparable to state-of-the-art values on the CIFAR-10 and ImageNet-10 datasets with simple $k$-means. We also verified IDFD loss works on multiple neural network structures, and our method is expected to be effective for various kinds of problems.

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

APPENDICES

## A  DATASETS AND EXPERIMENTAL SETUP

Five datasets were used to conduct experiments: **CIFAR-10** Krizhevsky et al. (2009), **CIFAR-100** Krizhevsky et al. (2009), **STL-10** Coates et al. (2011), **ImageNet-10** Deng et al. (2009), and **ImageNet-Dog** Deng et al. (2009). Table 3 lists the numbers of images, number of clusters, and image sizes of these datasets. Specifically, the training and testing sets of dataset STL-10 were jointly used in our experiments. Images from the three ImageNet subsets were resized to $96 \times 96 \times 3$.

Table 3: Image datasets used in experiments.

| Dataset | Images | Clusters | Image size |
|---|---|---|---|
| CIFAR-10 Krizhevsky et al. (2009) | 50,000 | 10 | $32 \times 32 \times 3$ |
| CIFAR-100 Krizhevsky et al. (2009) | 50,000 | 20 | $32 \times 32 \times 3$ |
| STL-10 Coates et al. (2011) | 13,000 | 10 | $96 \times 96 \times 3$ |
| Imagenet-10 Deng et al. (2009) | 13,000 | 10 | $96 \times 96 \times 3$ |
| Imagenet-Dog Deng et al. (2009) | 19,500 | 15 | $96 \times 96 \times 3$ |

We adopted ResNet He et al. (2016) as the neural network architecture in our main experiments. For simplicity, we used ResNet18, which according to our preliminary experiments yields sufficiently high performance. The same architecture was used for all datasets except the input layer. In accordance with the experimental settings of Wu et al. (2018), the dimension of latent feature vectors was set to $d = 128$, and a stochastic gradient descent optimizer with momentum $\beta = 0.9$ was used. The learning rate $lr$ was initialized to $0.03$, then gradually scaled down after the first 600 epochs using a coefficient of $0.1$ every 350 epochs. The total number of epochs was set to 2000, and the batch size was set to $B = 128$. Orthogonality constraint weights for IDFO were $\alpha = 10$ for CIFAR-10 and CIFAR-100 and $\alpha = 0.5$ for the STL-10 and ImageNet subsets. The weight for IDFO $\alpha$ was set according to the orders of magnitudes of the two losses $L_I$ and $L_{FO}$. For IDFD, the weight $\alpha$ was simply fixed at 1. In the main experiments, we set the default temperature parameter value $\tau = 1$ for ID(tuned), IDFO, and IDFD, and $\tau_2 = 2$ for IDFD.

## B  OPTIMAL SOLUTIONS OF CLUSTERING AND INSTANCE DISCRIMINATION

In Section 4.2.1, we concluded that minimizing $L_I$ under the condition that $\tau$ is large can reach a clustering-friendly solution. Details about the analysis and calculation was demonstrated by a two-dimensional toy model as follows.

Empirically, we observe that visually similar images tend to get similar assignment probabilities. Similar images can thus be projected to close locations in the latent space. This also motivated ID Wu et al. (2018). In the case of ID, similar images $x_i$ and $x_j$ yield respective highest probabilities $p_{ii}$ and $p_{jj}$, and also receive relatively high $p_{ij}$ and $p_{ji}$ values. This property can retain over the process of approximation to the optimal solution. Because instance-level discrimination tries to maximally scatter embedded features of instances over the unit sphere Wu et al. (2018), all representations are thus uniformly spread over the latent space with each representation relatively similar to its surroundings, we call this *uniform* case. We also consider another case that yields an optimal clustering solution where all samples from the same class are compacted to one point and $k$ clusters are uniformly spread over the space. We call this *compact* case. Figure 8 shows the representation distributions in the two cases. Because we normalize $v$, two-dimensional representations form a circle.

In the *uniform* case, $n$ representations are uniformly located on a circle with an angular interval of $\theta = 2\pi/n$, and the inner product between two neighboring representations is $\cos \theta$. Without loss of generality, we can start with an arbitrary point $v_i$ and orderly mark all samples as $v_{i+j}$. The cosine similarity between $v_i$ and $v_{i+j}$ can then be calculated by $v_{i+j}^T v_i = \cos j\theta$. Accordingly, the loss

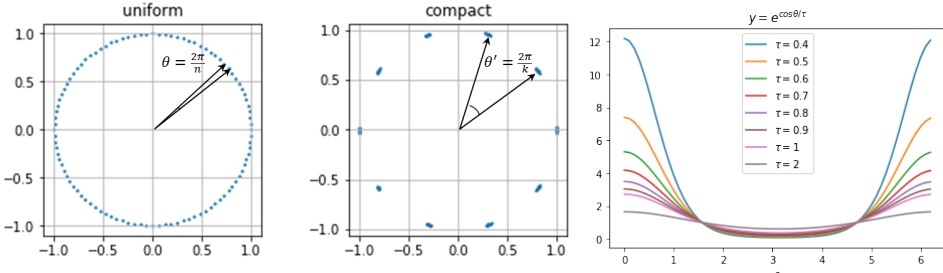

Figure 8: Two extreme cases of representation distributions over two-dimensional space. Left: *uniform*. Right: *compact*.

Figure 9: $\exp(\cos\theta/\tau)$ with different $\tau$ settings.

contributed by sample $i$ in the uniform case can be calculated as

$$L^i_{uniform} = -\log \frac{\exp(1/\tau)}{\sum_{m=0}^{n-1}\exp(\cos m\theta/\tau)} = -\log \frac{\frac{1}{n}\exp(1/\tau)}{\frac{1}{n}\sum_{m=0}^{n-1}\exp(\cos m\theta/\tau)}. \tag{11}$$

Similarly, in the *compact* case, $n/k$ data from the same class are exactly compacted to a point and $k$ corresponding points located on a circle at an angular interval of $\theta' = 2\pi/k$. The inner product between an arbitrary start sample $v_i$ and the $j$-th sample can be calculated as $v_i^T v_{i+j} = \cos l\theta'$, where $l = j \bmod n/k$. The probability of assigning $i$ to the cluster with $j$ becomes $p_{ij} = \frac{\exp(\cos\theta'/\tau)}{\sum_{c=0}^{k-1}\frac{n}{k}\exp(\cos c\theta'/\tau)}$. Accordingly, the loss contributed by sample $i$ in the compact case can be calculated as

$$L^i_{compact} = -\log \frac{\exp(1/\tau)}{\sum_{c=0}^{k-1}\frac{n}{k}\exp(\cos c\theta'/\tau)} = -\log \frac{\frac{1}{n}\exp(1/\tau)}{\frac{1}{k}\sum_{c=0}^{k-1}\exp(\cos c\theta'/\tau)}. \tag{12}$$

Comparing Eq. (11) and (12), we see that the difference between $L^i_{uniform}$ and $L^i_{compact}$ comes only from the denominator part of the logarithm. These are two discrete forms of the same integral $\int \exp(\cos\theta/\tau)d\theta$. Clearly, $L^i_{uniform}$ equals $L^i_{compact}$ when $k, n \to +\infty$. We therefore need to consider only the general case where $n$ is sufficiently large and $k \ll n$.

Figure 9 shows a plot of function values $\exp(\frac{\cos\theta}{\tau})$ with different $\tau$ settings over the domain $\theta \in [0, 2\pi]$. We can see that the curve becomes flatter as $\tau$ increases. A flat function $f$ means that for an arbitrary $(\theta, \theta')$ pair in its domain of definition, we have $f(\theta) \approx f(\theta')$. In this situation even $k \ll n$, the difference between the summations of these two discrete functions is not large. Accordingly, we can say $L^i_{compact}$ is approximate to $L^i_{uniform}$ for a large $\tau$. In other words, minimizing $L_I$ can approach the compact situation where same-class samples assemble and differing samples separate. Learning instance-level discrimination for clustering is therefore reasonable.

## C   EXTENDED EXPERIMENTS

In Section 4.2.3, we have reported some investigations of our method for practical use. Details about several important experiments are supplemented as follows.

### C.1   IMPACT OF NETWORK ARCHITECTURE

As Table 2 shows, IDFD can be applied to various networks, and the performance gaps between IDFD and ID(turned) on networks like ConvNet Wu et al. (2019) and VGG16 Simonyan & Zisserman (2014) are more significant than on ResNet He et al. (2016). We added the feature correlation matrix of VGG16 in Figure 10. IDFD on VGG16 obtained sparse correlations similar to the case of ResNet18 in Figure 7, while ID on VGG16 obtained denser and stronger correlations than ResNet18, presumably constructing redundant features that degraded clustering. In the case of VGG16, the feature decorrelation term $L_F$ exhibits a larger effect on clustering performance than that of ResNet.

Our proposed losses work on all network architectures, and we expect to introduce the losses to various networks that are suitable for individual problems.

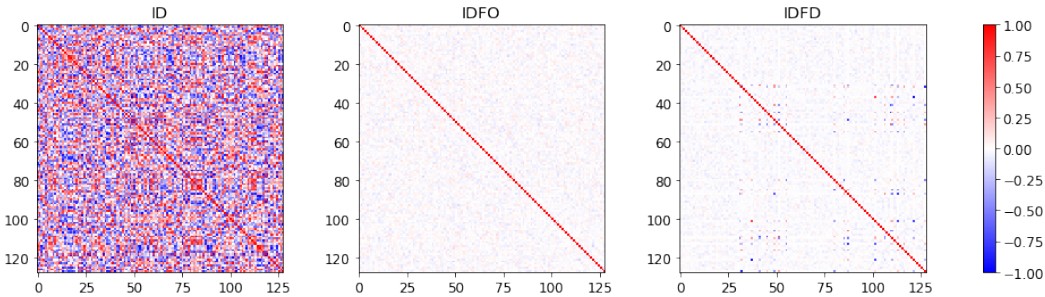

Figure 10: Feature correlation matrix learned by VGG16 on CIFAR-10.

## C.2 COMBINATION OF AUTOENCODER AND FEATURE DECORRELATION

In order to further confirm the cooperation effect of instance discrimination and feature decorrelation from the viewpoint of spectral clustering, a combination of autoencoder and feature decorrelation was evaluated in terms of clustering performance. Autoencoder has been verified by datasets such as handwritten digits to be an effective method for deep clustering. In this experiment, we used ConvNet Wu et al. (2019) for the autoencoder architecture and trained it on the CIFAR-10 dataset. We applied $k$-means to representations learned from autoencoder only and autoencoder combined with feature decorrelation, which are called AE and AEFD, respectively. According to our experiments, the ACC value of AE was $26.0\%$, and the ACC value of AEFD was $22.4\%$. Compared to the improvement from ID to IDFD (from $26.8\%$ to $42.0\%$ as shown in Table 2), we see that AE cannot benefit from FD as ID. This result again indicates that FD has a deep relation with ID as we analyzed in Section 3.

## C.3 IMPACT OF DATA AUGMENTATION

For reproduction of our results and practical use, we note that data augmentation (DA) has strong impact on the performance. DA is known to have impact on image classification and representation learning. Like in Wu et al. (2018), several generic and accepted techniques, such as cropping and grayscale, were used for data augmenting in this work. The details of the augmentation in the original code can be linked to Wu et al. (2018). In order to investigate the impact of DA, we conducted experiments on five datasets with and without DA and compared their clustering results. Table 4 shows the results. We can see that methods without DA suffered significant performance degradations for clustering, as well as for classification Chen et al. (2020). This reminds us not to ignore the effects of DA in practical use.

Table 4: Clustering results (%) with or without data augmentation on five datasets.

| Dataset | CIFAR-10 | | | CIFAR-100 | | | STL-10 | | | ImageNet-10 | | | ImageNet-Dog | | |
|---|---|---|---|---|---|---|---|---|---|---|---|---|---|---|---|
| Metric | ACC | NMI | ARI | ACC | NMI | ARI | ACC | NMI | ARI | ACC | NMI | ARI | ACC | NMI | ARI |
| ID W/O DA | 18.7 | 9.5 | 4.1 | 14.8 | 10.7 | 3.2 | 19.6 | 9.0 | 3.7 | 23.6 | 14.1 | 6.2 | 12.7 | 4.6 | 1.9 |
| IDFD W/O DA | 23.6 | 12.1 | 6.0 | 16.2 | 11.6 | 4.4 | 24.8 | 17.6 | 8.3 | 37.2 | 23.8 | 15.6 | 15.5 | 5.5 | 2.5 |
| ID With DA | 76.6 | 65.7 | 58.3 | 36.7 | 35.7 | 21.9 | 57.1 | 49.0 | 36.8 | 85.8 | 79.1 | 70.5 | 29.4 | 16.0 | 28.5 |
| IDFD With DA | 81.5 | 71.1 | 66.3 | 42.5 | 42.6 | 26.4 | 75.6 | 64.3 | 57.5 | 95.4 | 89.8 | 90.1 | 59.1 | 54.6 | 41.3 |

To further find out main factors affecting the performance, we also executed experiments by removing each technique used for DA. Take the example of CIFAR-10, techniques used for data augmentation include: *ColorJitter*, *RandomResizedCrop*, *RandomGrayscale*, and *RandomHorizontalFlip*. All these techniques are generic and easy to be implemented. They have been integrated into general deep learning frameworks such as PyTorch. According to our experimental results as shown in Figure 11, we find that *RandomResizedCrop*, *RandomGrayscale*, and *ColorJitter* have strong effect on image clustering.

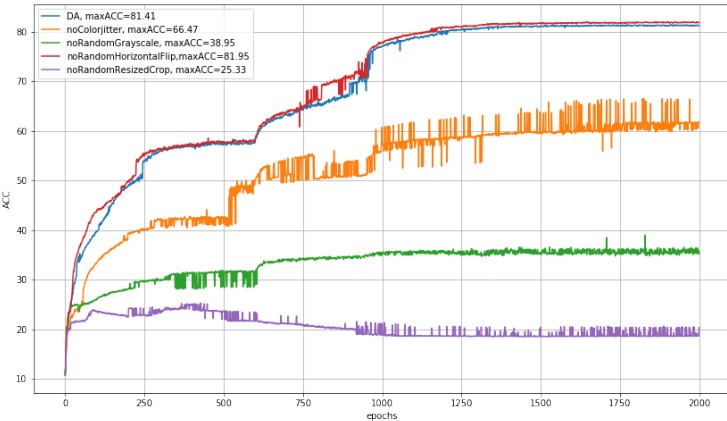

Figure 11: Effect of each technique used for DA on CIFAR-10.

For practice, we also applied IDFD to our private images produced by manufacturing process. Generic DA like above were used to these images. IDFD showed good performance on these images according to our experiments. This indicates that our method can be simply applied to practical images. For other types of data such as text and time series, corresponding data augmentation techniques are needed to cooperate with our method.

