# OpenReview forum: "Clustering-friendly Representation Learning via Instance Discrimination and Feature Decorrelation"
_ICLR.cc/2021/Conference — ICLR 2021 Poster_

### Official Review · AnonReviewer4 · 2020-10-29
**This paper proposes two interesting contributions to the 'deep clustering' literature and demonstrates the benefit experimentally.**

**Rating:** 6
**Confidence:** 3

**Review:**

One of the main contributions is this idea of feature decorrelation where they encourage the representation features to be independent / orthogonal.
The other is instance discrimination. This aims to capture the similarity between individual data points. Both of these are interesting contributions to the field of 'deep clustering'.


Besides the stated contributions, I thought there were a number of other positive aspects of this.
	A) I thought that the spectral clustering connection was nice and I am glad the authors included it.
	B) The evaluation is fairly detailed. I particularly appreciate the fact that the authors used datasets that are somewhat larger than often used in the literature (MNIST and CIFAR-10 vs CIFAR-100 and ImageNet-10). The inclusion of the study of the temperature parameter also helped clarify a few questions I had when reading it.
	C) Finally, the evaluation clearly shows the benefit of their contributions in terms of performance.

There are a number of questions I have with the work as is.
	A) Given the two methods proposed,  IDFO, IDFD, neither of which outperforms the other on all tasks, and given this is unsupervised learning, how does one know which method to use?
	B) Why was the alpha parameter set to 1 for IDFD? How does one know what to set this to for different datasets? If it's always 1, why is it included at all?  This is particularly important to understand in unsupervised settings.
	C) The impact of data augmentation is discussed in the supplementary but this is stated as being extremely important to the performance of the model. It is unclear to me whether the results in the main text include the augmentation process?  If so, then given this, I think it should be stated in the main text as it has an effect on both instance discrimination and feature decorrelation considering the addition of augmented images.  The results in supplementary Table 4 include KNN and don't match up with the main results in the main text which further confused me.
	D) I was left wondering how well this method works on non-image data? Other works in the literature have explored this.
	E) For Fig. 2 is this ACC calculated on the validation set or test set?
	F) What were the effects of resizing the ImageNet images? Can this model handle larger images, and if so, how does this effect performance?

Minor
	A) References are badly formatted in Table 3.


Overall, my questions above notwithstanding, I think this is an interesting contribution which shows the benefit of instance discrimination and feature decorrelation for deep clustering.

---

> ### Author Response · Authors · 2020-11-14
> **Reply to Reviewer4**
>
> Thank you very much for your comments. Please find below our responses to each of your comments.
> 1.	"Given the two methods proposed, IDFO, IDFD, neither of which outperforms the other on all tasks, and given this is unsupervised learning, how does one know which method to use?"
> One of the advantages of IDFD compared with IDFO is performance stability. We plotted the accuracy values of IDFD and IDFO over the whole learning process in Figure 2. The performance of IDFO is higher than the other methods in earlier epochs. However, the accuracy widely ﬂuctuated over the learning process and dropped in later epochs. Totally, we recommend IDFD for its performance and stability.
> 2.	"Why was the alpha parameter set to 1 for IDFD? How does one know what to set this to for different datasets? If it's always 1, why is it included at all? This is particularly important to understand in unsupervised settings."
> According to the definition of ID and FD, two loss terms are resultantly of same order, and the $\alpha$ parameter was simply fixed to1 in our experiments. For the general formulation of loss function with two terms, we have prepared a weight term alpha. We think it is not easy to determine a certain value of alpha for different datasets, as for the weights of many other losses.
> 3.	"It is unclear to me whether the results in the main text include the augmentation process? If so, then given this, I think it should be stated in the main text…"
> Yes, the augmentation process was included in the main results. We will state that in the main text of next revision.
> 4.	"The results in supplementary Table 4 include KNN and don't match up with the main results in the main text which further confused me."
> Original instance discriminative method ID (original) was only evaluated by KNN metric but clustering metrics ACC, NMI, and ARI. In order to illustrate the impacts of ID (original) from data augmentation, we reserved the KNN results in our paper. We can delete it to improve the readability and clarity.
>  Difference between the results of Table 1 and 4 is due to different values for parameter $\tau_2$. We will  execute these experiments with the same parameters in the revision.
> 5.	"I was left wondering how well this method works on non-image data? Other works in the literature have explored this."
> Up to now, we mainly attempted our method on image data including both open and our private data. It yielded good performance as expected. Further experiments on other data such as time-series data are undergoing.
> 6.	"For Fig. 2 is this ACC calculated on the validation set or test set?"
> ACC in Fig.2 is calculated on the train set. Because our method does not use the label information in training, we do not prepare train/valid/test sets and simply learn the representation from a set of data and evaluate our method with the set. We used the train set for training and evaluation rather than the whole dataset for the consistency with previous works.
> 7.	"What were the effects of resizing the ImageNet images? Can this model handle larger images, and if so, how does this effect performance?"
> One effect of resizing the ImageNet images is reducing the computation cost and memory consumption. Compared with CIFR-10, ImageNet consists of bigger images. We referred to the previous work DCCM to resize images. We also executed the experiment on a smaller size 32x32 and got 87.6% ACC values on ImageNet-10, which is reduced by 7% on 96x96. From this result, we expect that the performance on larger images will be improved.
> 8.	“References are badly formatted in Table 3.”
> We will fix it in the revision.
>
> Thank you very much.

---

### Official Review · AnonReviewer2 · 2020-10-29
**Paper 1734 review**

**Rating:** 7
**Confidence:** 5

**Review:**

This paper proposes a clustering-friendly representation learning method using instance discrimination and feature decorrelation. Instance discrimination loss and feature decorrelation loss are combined to optimize the network. The paper is well qritten and experimental results are good. I have some questions about this paper:
1. There is no ablation analysis about the two loss terms in Eq.(6). What about the contributions of the two loss terms?
2. What is the motivation of Eq.(3)? I.e., why the "=" holds between the second and third expressions?

---

> ### Author Response · Authors · 2020-11-13
> **Reply to Reviewer2**
>
> Thank you very much. Please find below our responses to each of your questions.
> 1.	"There is no ablation analysis about the two loss terms in Eq.(6). What about the contributions of the two loss terms?"
> The results of ID(tuned) in Table 1 show the clustering performance only with the first term ID. The gaps between results of ID(tuned) and IDFD indicate the contribution of the second term FD.
>
> 2.	"What is the motivation of Eq.(3)? I.e., why the "=" holds between the second and third expressions?"
> In our definition and notation, $v$ and $f$ have a relationship of $v^T=f$, which derives the equation of the second and third expressions in Eq. (3).

---

### Official Review · AnonReviewer1 · 2020-10-29
**A good submission that aims at a valuable and fundamental machine learning task that shows some improvement.**

**Rating:** 7
**Confidence:** 4

**Review:**

The authors proposed an improved deep-learning-based representation learning method that provides more efficient features for clustering analysis.
(1) According to the comparison experiments on several widely used datasets,  the integration of a softmax-formulated orthogonal constraint is able to provide more stable latent feature representation.
(2) As far as know, the widely-used deep clustering methods used to alternatively optimize the feature representation model parameters and update the anchors that provided by clustering method such as k-means, I am wondering if the proposed method in this study could integrate the two steps in a real end-to-end fashion.
(3) I was deeply impressed by the far above state-of-the-art values of evaluation metric of this proposed representation learning method. Although the authors provide some distribution illustrations of latent features on CIFAR-10 dataset, what about the visualization on the ImageNet-10? Besides, adding some 'real' visualization results existing in the original image space rather than the latent space could help to illustrate if the proposed method could mine visually meaningful concepts from the view of visual contents.

---

> ### Author Response · Authors · 2020-11-13
> **Reply to Reviewer1**
>
> Thank you very much for your comments. Please find below our responses to each of your comments.
>
> 1.	“I am wondering if the proposed method in this study could integrate the two steps in a real end-to-end fashion.”
> In this work, we focus on the task of representation learning which is essential for the performance of clustering. The insight on IDFD’s relation with spectral clustering, which performs construction of graph and its projection to lower dimension before clustering, is another motivation to focus on the representation learning.  Our method can be integrated with other clustering methods (including deep clustering methods).
>
> 2.	“what about the visualization on the ImageNet-10?”
> We only gave the visualization on CIFAR-10 due to the limitation of space. The visualization on the ImageNet-10 can be shown in the next revision.
>
> 	“adding some 'real' visualization results existing in the original image space rather than the latent space”
> It is not easy to visualize the high dimensional data in the original space. We can show several samples of original images corresponding to the points on the latent space in the next revision.

---

### Public Comment · ~Wouter_Van_Gansbeke1 · 2020-11-10
**Missing relevant prior work**

Dear authors and reviewers,

We would like to point out an important competing method [A] that was omitted from the comparison in this paper. The referred work also considers the problem of unsupervised image classification or image clustering, and was accepted as a conference paper at ECCV'20. This happened well before the ICLR'21 submission deadline. We would like to point out the following items:

1)[A] is the current state-of-the-art in unsupervised image classification and outperforms the method under review. More specifically the following results are obtained on CIFAR-10: 88.3 (+5.5%), STL-10: 80.9 (+5.3%) and CIFAR100-20: 50.7 (+8.2%). Furthermore, the method in [A] reported results on the full ImageNet dataset with 1000 classes. We find it unfortunate that [A] was left out from the state-of-the-art comparison. Particularly because the code was made publicly available, and can also be found on the relevant leaderboards of the well-known ‘Papers With Code’ Platform. (https://paperswithcode.com/paper/learning-to-classify-images-without-labels).

2)The idea of using an instance discrimination task for the purpose of image clustering was already provided by [A]. In particular, section 2.1 in [A] motivated the use of the instance discrimination task to learn feature representations that are well-suited for the down-stream task of semantic clustering.

3)Finally, we also noticed that the authors excluded the prior state-of-the-art [B] (before [A]).

[A] Van Gansbeke, Wouter, et al. "Scan: Learning to classify images without labels." ECCV. 2020.
[B] Xu, Ji, et al. "Invariant information clustering for unsupervised image classification and segmentation."ICCV. 2019.

---

> ### Author Response · Authors · 2020-11-13
> **Reply to Missing Relevant Prior Work**
>
> Thank you for your information. We realize that we have omitted the previous works. We will add the description about [A] and [B], in which [A] has achieved the state-of-the-art clustering accuracy, and the comparison with our work in our revised paper. In addition to that, we would like to clarify several points by responding to your comments.
> 1.	Both [A] and IDFD classify data without labels but they are inherently different. [A] focuses on the clustering phase after representation learning, while IDFD focuses on the representation learning phase. To clarify the effect of representation learning, IDFD uses simple k-means algorithm in the clustering phase. We consider [A] and IDFD are compatible and can be combined to get higher performance in the whole clustering task.
> 2.	For the comment “The idea of using an instance discrimination task for the purpose of image clustering was already provided by [A]”, we need to note that we do not just use the original instance discrimination for clustering in our work. We analyzed why and on what conditions the instance discrimination can be used for clustering. Though [A] mentioned ‘instance discrimination task to learn feature representations that are well-suited for the down-stream task of semantic clustering’, the results in Table 2 of [A] showed the original instance discrimination method only got about 52% accuracy on CIFAR-10 which is lower than our 81%. The significant gap also shows the contribution of IDFD on the representation learning phase. We would like to further note that IDFD is demonstrated to be analogous to the classical spectral clustering, which has been theoretically and experimentally investigated, and known to outperform other traditional clustering methods.
> 3.	About “the prior state-of-the-art [B]“, which demonstrated completive performance with DCCM referred in Table 1, we would like to list the results in our revised paper.
>
> Thank you for your information again.

---

> > ### Public Comment · ~Wouter_Van_Gansbeke1 · 2020-11-14
> > **Reply to Authors**
> >
> > We would like to thank the authors for addressing our comments on the paper.
> >
> > The main concern we expressed was the state-of-the-art comparison. In particular, two important works [A,B] (prior and current state-of-the-art) were excluded from the comparison. The authors promised to include both works in the camera-ready version of their paper, in which case this issue is addressed. Still, we would like to emphasize that the state-of-the-art claims should be expressed a bit more carefully.
> >
> > We agree with the second point of the authors. Both [A,B] avoided the use of K-Means, and directly predicted the cluster assignments using the network. The motivation for this was that K-Means can easily lead to cluster degeneracy. Taking this into consideration, the main contribution of this paper is to learn a feature space that can be directly clustered using K-Means. This marks a large deviation from prior works [A,B]. Experimentally, this paper shows that applying K-Means to the feature space outperforms the prior state-of-the-art [B], or can perform on par with the current state-of-the-art [A] in some cases. We believe it would be useful to add this discussion to the paper.

---

### Author Response · Authors · 2020-11-20
**Reply letter to reviewers: a summary of our revision (imported: 20 Nov 2020)**

We really appreciate your time and valuable comments. We have carefully revised our paper, taking into consideration reviewers’ and public comments. The main revisions are summarized as follows.

1. We added the description about two previous works, one of which has achieved the state-of-the art clustering accuracy, and compared them with our work in the revision. (Section 2 in page 3, Table 1 in page 7.)
2. We added representation visualization on the ImageNet-10 and plotted sample images corresponding to points near the border between clusters. (Section 4.2.2 Figure 6 in page 8)
3. We re-executed experiments in Appendix C.3 and updated the results in Table 4 by setting parameter $\tau_2$ same with the main experiments in Section 4.
4. We deleted the results of KNN metric, which is often used in representation learning for supervised classification, to improve the readability and clarity in Appendix C.3 Table 4.
5. We added the description to clarify that the data augmentation is used in main results. (Section 4 in page 6.)

Other minor revisions such as reference format are included responding to comments.

---

### Decision · Program_Chairs · 2021-01-07
**Final Decision**

**Decision:**

Accept (Poster)

**Comment:**

This paper received mostly positive reviews. The reviewers praised the strong performance when compared with previous work.
Also, the evaluation clearly shows the benefit of the proposed contributions in terms of performance.
Most concerns raised by reviewers were properly addressed in the rebuttal.

Lack of comparison to several previous works has been noted in a comment, but the authors clarified this concern, stating that the current work is a “large deviation from prior works”. The authors promised to include the missing references into the comparison.

Given the reviews, comments, and author's answers, I suggest acceptance.